# From Benznidazole to New Drugs: Nanotechnology Contribution in Chagas Disease

**DOI:** 10.3390/ijms241813778

**Published:** 2023-09-07

**Authors:** Daniele Cavalcante Gomes, Thayse Silva Medeiros, Eron Lincoln Alves Pereira, João Felipe Oliveira da Silva, Johny W. de Freitas Oliveira, Matheus de Freitas Fernandes-Pedrosa, Marcelo de Sousa da Silva, Arnóbio Antônio da Silva-Júnior

**Affiliations:** 1Laboratory of Pharmaceutical Technology and Biotechnology, Department of Pharmacy, Federal University of Rio Grande do Norte-UFRN, Natal 59012-570, Brazil; danicavalcanteg@gmail.com (D.C.G.); thaysesmfarma@gmail.com (T.S.M.); eronlincolnap@gmail.com (E.L.A.P.); joao.silva.706@ufrn.edu.br (J.F.O.d.S.); matheus.pedrosa@ufrn.br (M.d.F.F.-P.); 2Immunoparasitology Laboratory, Department of Clinical and Toxicological Analysis, Centre of Health Sciences, Federal University of Rio Grande do Norte-UFRN, Natal 59012-570, Brazil; johny3355@hotmail.com (J.W.d.F.O.); mssilva.ufrn@gmail.com (M.d.S.d.S.)

**Keywords:** neglected diseases, *Trypanosoma cruzi*, nanotechnology, drug delivery systems, drug targeting

## Abstract

Chagas disease is a neglected tropical disease caused by the protozoan *Trypanosoma cruzi*. Benznidazole and nifurtimox are the two approved drugs for their treatment, but both drugs present side effects and efficacy problems, especially in the chronic phase of this disease. Therefore, new molecules have been tested with promising results aiming for strategic targeting action against *T. cruzi*. Several studies involve in vitro screening, but a considerable number of in vivo studies describe drug bioavailability increment, drug stability, toxicity assessment, and mainly the efficacy of new drugs and formulations. In this context, new drug delivery systems, such as nanotechnology systems, have been developed for these purposes. Some nanocarriers are able to interact with the immune system of the vertebrate host, modulating the immune response to the elimination of pathogenic microorganisms. In this overview of nanotechnology-based delivery strategies for established and new antichagasic agents, different strategies, and limitations of a wide class of nanocarriers are explored, as new perspectives in the treatment and monitoring of Chagas disease.

## 1. Introduction

Chagas disease (CD), also known as American trypanosomiasis, is a neglected tropical disease (NTD), with an estimated 6–7 million people infected worldwide, with 12,000 deaths per year. This disease is considered the parasitic disease with the greatest socioeconomic impact in Latin America. It is present in 21 countries, with an annual incidence of 30,000 new cases on average, and is an important public health problem (PAHO/WHO, 2020). Since the past decade, the intense migration of infected individuals from endemic areas to North America, Europe, and Asia has led CD to be considered a global threat [1,2,3].

Currently, there are only two approved drugs for CD treatment in the world. The treatment regimen with nifurtimox (NFX) or benznidazole (BNZ) is long and involves several side effects that lead to patient compliance problems. The efficacy of BNZ, as well as biopharmaceutical behavior, are well established, which explains its use as the main pharmacological treatment. Different studies are performed to solve many drawbacks, such as both drug solubility and specific drug bioavailability [4,5,6].

In the treatment in the acute phase of infection, in which the circulating trypomastigote form is present, both BNZ and NFX are effective. However, in the chronic phase, in which the non-circulating amastigote form of *T. cruzi* is present in myocardial tissue, their benefits are limited, with a greater variation in the efficacy, relating to the tropism of the parasite in different tissues and immunological condition of the patient. Furthermore, the genetic heterogeneity of parasites is complex, and some strains present drug resistance [7,8,9,10,11]

Due to the limitations regarding the use of BNZ and NFX, new synthetic or natural molecules with anti-*T. cruzi* activity have been identified and evaluated. However, these approved drugs present similar problems to that identified in BNZ and NFX [12]. Occasionally, an excellent in vitro efficacy result is not reproduced when tested in vivo when important and complex biological barriers are present [13,14,15].

Nanotechnology approaches have shown important results in the search for strategies that can enhance the performance of BNZ, NFX, or new drugs against *T. cruzi*. These studies include efficacy, toxicity, stability, and drug bioavailability. However, the greater concern remains about drug targeting and resistance mechanisms of the parasite in conjunction with a prolonged release compared to conventional drugs [16,17]. In this review, different antichagasic drug delivery systems are presented and discussed as their in vitro or/and in vivo performance. To this end, studies available on Web of Science, PubMed, or Scopus were considered, considering the keywords (“American Trypanosomiasis” OR “Chagas Disease” OR “*Trypanosoma cruzi*” OR “South American Trypanosomiasis” OR benznidazole OR nifurtimox) AND (“nanoparticles OR nanotechnology OR “drug delivery systems” OR microparticles OR nanoemulsions OR nanosystems OR nanostructures OR nanomedicine OR cyclodextrins OR “solid dispersions” OR “Metal Organic Frameworks” OR “inclusion complexes”). This search resulted in 40 studies, most of the investigations were authored by Brazilian researchers (21 studies, 53%), followed by Argentina (11 studies, 28%), Chile (03 studies 8%), Spain (02 studies, 5%), and Mexico, Paraguay, Belgium (01 study each, 3%) as shown in Figure 1. Additionally, six patents from Brazil involving nanocarriers for treatment were also identified, as shown in Table 1. According to The Lancet Global Health, a study by Belen Pedrique and colleagues showed that in the last 10 years, 850 new therapeutic products have been registered, of which only 36 (4%) have indications for neglected diseases. Among them, only one pediatric formulation of benznidazole is described to treat Chagas disease [18]. Latin American countries have sparsely researched new drug delivery systems to treat CD, but there is still an urgent need for the scientific community to address greater efforts to this seriously neglected public health problem.

The word cloud recorded the types of drug delivery systems, drugs, mechanism of action, and type of study revealing interesting and more frequent keywords in these studies. As presented in Figure 2, liposomes, polymeric nanoparticles, and nanoemulsions are most investigated for this purpose. BNZ remains the most investigated drug, with 17 in vitro and 23 in vivo studies.

In this review, firstly, Chagas disease is initially present as a challenge for therapeutics. This is mainly connected with the parasite cycle and how the different evaluative forms of *T. cruzi* induce different clinical stages of this disease. Different nanocarriers are presented based on the target of drug action (free radical formation, sterol biosynthesis inhibitors, RNA strand inhibition, purine salvage inhibitor, enzyme inhibition, autophagy induction, tubulin assembly inhibition, immunomodulation). We believe that this approach can help us to better understand and reveal possible new drug-targeting strategies to be used against *T. cruzi*.

## 2. Chagas Disease

Chagas disease is an anthropozoonosis caused by the hemoflagellate protozoan *T. cruzi*. It is naturally transmitted to humans by the feces of hematophagous triatomine vectors. But other routes of transmission include blood transfusion [28] or contaminated organ transplantation [29], ingestion of contaminated food [30], accidental laboratory infections [31], and congenital transmission from infected mothers to newborns [32,33].

Three predominant forms are present in biological cycle of *T. cruzi* (i) epimastigote, (ii) trypomastigote, and (iii) amastigote. The epimastigote form is located in the midgut of the insect proliferating by binary multiplication, the morphological characteristic of this form is defined by the kinetoplast position before the core resulting in a different flagella development. The infectious form, trypomastigote, is found in mammalian blood (trypomastigote bloodstream) and in the hindgut of triatomine (metacyclic form), the forms present in the kinetoplast after the core and flagella formation promote a different movement for the parasite. Finally, the amastigote form, a circular form present inside mammalian cells characterization the chronic stage when located in tropism tissues, this form is replicative too, and highly resistant to the immune mechanisms of the host [34].

In Latin America, where Chagas disease is endemic, a remarkable variation in the distribution of the pathogenesis and morbidity of this disease is observed [3]. The origins of this geographical heterogeneity in clinical outcomes, as well as the mechanisms leading to the development of the different clinical forms of Chagas disease, are still uncertain and poorly understood [35]. It is proposed that the complexity in the pathogenesis of Chagas disease is probably determined by several distinct factors, such as the gene diversity of the parasite, the immune status of the host and environmental factors, and the interaction between all these elements [11,36].

CD usually presents in two distinct phases: (i) acute phase and (ii) chronic phase, which in turn can be categorized into indeterminate, cardiac, or digestive forms, each with different clinical manifestations [3]. During the acute phase, it is common to detect the parasite microscopically. Symptoms in this phase are usually mild and non-specific, including fever, malaise, hepatosplenomegaly, and atypical lymphocytosis [35,37,38,39]. In some rare cases, the presence of a nodule on the skin, known as a chagoma or Romaña sign, may be noted, which indicates the site of parasite inoculation [37].

After the acute phase of CD, in which the cell-mediated immune response controls parasite replication and symptoms disappear spontaneously, parasitemia usually disappears within 4–8 weeks [40]. Without an effective etiologic treatment, the disease reaches the chronic phase, which sets in gradually. In this phase, most chronic patients are asymptomatic and present an indeterminate clinical form of the disease and may remain so indefinitely [41].

However, after an asymptomatic period of 10–30 years, approximately 30–40% of chronically infected individuals develop cardiac abnormalities, while about 10% of them develop the digestive or neurological form of the disease [40,42,43]. The digestive form of CD is a result of neuronal destruction in the enteric nervous system, leading to peristaltic dysfunction and consequently dysphagia. Hypomotility of the digestive system leads to dilatation, especially of the esophagus and colon, causing conditions known as megaesophagus and megacolon [44,45].

Chronic Chagas cardiomyopathy is one of the most debilitating clinical forms of the disease, affecting about one-third of infected individuals [46,47]. This condition can cause serious consequences such as sudden death, cardiac arrhythmias, heart failure, and thromboembolism [48]. Therefore, CD is considered a disabling disease and responsible for the highest morbidity and mortality among parasitic diseases [49], but the mechanisms or determinants responsible for the development of cardiomyopathy are unclear [50].

Currently, following the WHO recommendation, the treatment of CD is based on two nitroheterocyclic compounds: benznidazole and nifurtimox; this approach has been used since the 1970s [51]. Both act as prodrugs and need to be activated by nitroreductases present in *T. cruzi* itself to exert their cytotoxic effects. However, their mechanisms of action have not yet been fully elucidated [52]. In addition, they have limited efficacy only in the acute phase of the disease, as prolonged treatment has been associated with severe side effects and parasite resistance [8,53,54]. Among the most common side effects are anorexia, nausea, headache, vomiting, manic symptoms, seizures, paresthesia, and dermatitis, leading to discontinuation of therapy in about 10–30% of treated patients [36].

Thus, the use of technological alternatives is highly desirable, aiming at possible improvements in the treatment of Chagas disease. These alternatives include the development of new formulations capable of increasing drug bioavailability and improving biopharmaceutical properties in the search for new molecules with potential trypanocidal activity. In addition, the search for these alternatives may contribute to the reduction of side effects associated with the current treatment and potentially increase therapeutic success for patients affected by the disease also in the chronic phase [55,56,57].

## 3. Nanocarriers: General Aspects

The fight against *T. cruzi* infection is a major challenge since the intracellular spread of the parasite is difficult to control. With low drug concentrations reaching the intracellular form, due to the limited biopharmaceutical properties of the drugs of choice for the treatment of Chagas disease, colloidal drug delivery systems gain strength due to their nano size, as they have the ability to increase intracellular delivery. In addition, they can increase the stability, activity, and kinetics of actives in the target cellular environment. Thus, these nanotechnology systems have proven to be very useful and relevant for the pharmaceutical industry [58,59,60,61].

Nanotechnology can be characterized as the science that aims to develop systems and materials at a nanometer scale, with sizes between 1 and 1000 nm. It was first mentioned in the late 1950s by physicist Richard P. Feynman, but it was not until 1974 that the term nanotechnology was introduced by Norio Taniguchi. In 1991, the term nanomedicine was publicized and was related to research into the possibilities of controlling and manipulating cellular processes, for example by targeted delivery of drugs and active substances [62,63,64,65]. Since then, this technology has been investigated for the treatment of various diseases.

Due to the lack of new drugs for CD, nanotechnology can be a great tool for the treatment of this neglected disease. This drug delivery process in nanomedicine can be realized from different types of nanosystems, such as polymeric nano- and microparticles, nano- or microemulsions, lipid nanoparticles, inorganic nanoparticles, solid dispersions, liposomes, micelles or metal–organic structures.

Scientifically recognized for over 50 years, liposomes can be defined as spherical, self-assembled systems composed of one or more layers of amphiphilic phospholipids and an aqueous core. They are often used in drug delivery systems because of their high biocompatibility, simplicity of manufacture, and chemical variability. They are biodegradable, biocompatible, non-immunogenic, and highly versatile for investigational, analytical, and therapeutic applications [66]. An important distinguishing feature of liposomes is their ability to simultaneously transport hydrophilic and lipophilic molecules [67,68,69,70]. Currently, there are some nano drugs based on liposome technology, such as Doxil^®^, Ambisome^®^, Onivyde™, and others [71]. Starting from liposomes, lipid nanoparticles are subdivided into solid lipid nanoparticles (SLN) or nanostructured lipid carriers (NLC) [72]. Where SLN is formed by solid lipids at room and body temperature, and NLC, the second generation of lipid nanoparticles, is composed of a liquid lipid in addition to the solid lipid, reducing the crystallinity of the particle matrix and increasing the stability time of the system with the incorporation of the active [73]. Another type of lipid nanosystem is the nanoemulsion, characterized as the colloidal dispersion of two immiscible liquids, where one of them composes the dispersed internal phase and the other the external phase, forming droplets of average diameter on the nanometric scale [74,75]. These systems are widely used for drug delivery since the existence of polar and non-polar phases at the interface of the system promotes a high ability to incorporate molecules with different characteristics in terms of solubility [76].

Polymeric nanoparticles can be prepared from synthetic or natural polymers. They comprise one of the most popular types of nanosystems due to their wide application, easy production, and high availability of biocompatible and biodegradable polymers. They have advantages such as taking molecules to the target site due to their ability to overcome biological barriers in various diseases [77,78,79]. Also consisting of polymers, solid dispersions are systems in which drugs with low aqueous solubility and consequent limited oral bioavailability are dissolved at the molecular level in an amorphous polymeric vehicle, where the solubility of an active substance is enhanced by disrupting its crystalline network to achieve a high-energy amorphous state [80,81]. Furthermore, polymeric micelles are nanoscale colloidal structures formed by the self-association of amphiphilic compounds or surfactants under a given temperature and concentration. Their structure consists of the core architecture, composed of a hydrophobic core that can encapsulate drugs, improving their solubility and reducing their toxicity [82,83].

In addition to organic components, nanosystems can also be structured from inorganic components, such as inorganic nanoparticles, which are spherical structures of easy functionalization and broad physiological applicability and can be prepared in three ways, through physical, chemical, or biological methods. They can consist mainly of metals, metal oxides, metal alloys, and semiconductors, such as gold, iron oxide, silver, zinc oxide, silica, and silicon dioxide, among others. In addition, they can be functionalized or hybridized with organic compounds such as polymers [84,85,86]. We can also mention metal–organic frameworks (MOFs) which comprise a class of porous crystalline solid materials that are self-assembled by metal ions and ligands. They are very promising in the encapsulation, storage, and release of functional biomolecules due to their low density, adjustable porous structures, high specific surface areas, ordered crystalline structure, and compositions controllable at the molecular level [87,88,89].

These nanosystems have been shown to enhance the therapeutic effect of different drugs for the treatment of diseases such as cancer [90,91], bacterial infectious diseases [92,93], inflammatory processes [94], and especially neglected tropical diseases such as dengue [95,96], African trypanosomiasis [97], tuberculosis [98], human schistosomiasis [99], and cutaneous [100,101] or visceral leishmaniasis [102]. These different studies demonstrate that they are able to reduce the toxicity of various bioactive molecules, as well as improve their biopharmaceutical and pharmacokinetic aspects [59]. Therefore, this review brings an overview of all the studies reported in the literature to date that have applied nanotechnology as a drug delivery system of different molecules, with the aim of fighting *T. cruzi* infection, as a smart, innovative, and effective alternative for the potential treatment of CD.

## 4. Therapeutic Targets for Chagas Disease

New molecules are being investigated for their antichagasic activity. Figure 3 summarizes the main compounds under investigation. In the following topics, we highlight their potential targets for action on *T. cruzi*, as well as the type of nanosystem used to incorporate it (Figure 4).

### 4.1. Action on Ergosterol

Ergosterol is an endogenous structural lipid important for the growth and survival of *Trypanosoma cruzi*. Similar to cholesterol in mammals, this sterol plays a crucial role in the stability, fluidity, and permeability of biological membranes. In addition to modulating the activity of membrane-bound enzymes and ion channels, it demonstrates its role in the formation of viable membranes and in different regulatory processes that are essential for parasite development and division. Unlike mammals that accumulate cholesterol, protozoa such as *T. cruzi* are not able to accumulate ergosterol. For this reason, blocking ergosterol production is considered lethal for the parasite [103,104].

They resemble fungi in terms of the cellular composition of steroids and their biosynthesis; thus, triazoles used clinically as antifungal agents may act through inhibition of C14-α-demethylase (CYP51) dependent cytochrome P-450 in *T. cruzi*, generating the accumulation of 14-α-methylesterol, a precursor of toxic methylated sterols, which also promote parasite growth arrest and deleterious changes in membrane permeability, including electron transport, and can cause parasite death [104].

The experimental azole drug, D0870, was the first to show a cure for *T. cruzi*-infected mice in the chronic phase of the disease [105]. In 2001, Molina and co-workers conducted a study in which D0870, itraconazole, and ketoconazole were incorporated alone into polyethylene glycol-poly(lactic acid) (PEG-PLA) nanospheres and monodisperse systems ranging in size from 100 to 200 nm were obtained. The incorporation efficiency was 90, 87, and 92% for DO870, itraconazole, and ketoconazole, respectively. For the in vivo assays, Swiss albino rats were infected intraperitoneally with the CL and Y strains in the trypomastigote form. The treatment was carried out for 30 consecutive days intravenously, administering a nanoparticulate system with DO870. It was observed that for doses of 1.5 and 3 mg/kg/day, there was a 70 and 90% cure rate for CL, respectively, and a 60% cure rate for the 3 mg/kg/day dose in strain Y. The results found regarding the cure rate were similar when compared to the free oral drug, which showed an 80% cure rate at a dose of 5 mg/kg/day for the CL strain. As for the Y strain, the results were superior when compared to orally via BNZ, which showed 47% cure at a dose of 100 mg/kg/day. The study also highlighted that no anti-*T. cruzi* activity was observed for itraconazole and ketoconazole [106]. However, D0870 had its studies stopped in the clinical phase of the trials (for fungal infections) due to undesirable side effects [107,108].

Another triazole antifungal studied was ravuconazole (RAV) which demonstrated efficacy limitations in in vivo studies in murine models upon oral administration, possibly due to the expected pharmacokinetic profile for a Biopharmaceutical Classification System (BCS) class II drug [109]. Self-emulsifying drug delivery systems (SEDDSs) have often demonstrated the potential to improve the oral bioavailability of lipophilic drugs, particularly class II drugs [110]. Thus, SEDDSs composed of Miglyol^®^ 810N, Labrasol^®^, and Epikuron^®^ were produced with RAV [111]. Nanogot particles with a diameter of 247 nm and uniform size distribution were obtained, with a polydispersity index (PdI) below 0.4. In in vitro assay in H9c2 cells to test trypanocidal activity on the amastigote form of strain Y, it was observed that RAV-SEDDS increased the inhibition activity against amastigotes ~1.8-fold compared to free RAV at doses equivalent to IC50 (0.1 nM), showing a clear improvement in the therapeutic efficacy of RAV against the intracellular form of the parasite. Furthermore, in vivo toxicity tests using Swiss mice, treated with RAV SEDDS (10%, *v*/*v*) for 20 days, showed no weight loss for animals treated with RAV-SEDDS or free RAV, indicating no additional toxicity of the drug. The authors concluded that RAV in the SEDDS pharmaceutical form is a strategy that deserves further in vivo experiments and preclinical studies as a potential treatment for human *T. cruzi* infections.

Ramos-Ligonio, Lõpez-Monteon, and Trigos (2012) developed hybrid systems called metal–organic frameworks (MOFs) using ergosterol peroxide (EP), which is a natural compound abundant in many species and with a structure similar to ergosterol, which can be a good anti-*T. cruzi* strategy due to its selectivity for the plasma membrane [112]. In this study, the zinc (II) polymer formed by two organic ligands, 4,4’-bipyridyl and acetate, was used, resulting in particles with sizes less than 100 nm and a surface charge of +31 mV. MOFs-EP showed an IC50 value of 4.81 ng/mL and 3.0 ng/mL for 24 and 48 h, respectively. The inhibition in parasite growth in the presence of MOFs-EP was comparable to that observed in the culture of parasites that interacted with NFX. In the end, it was observed that the use of MOFs-EP has an inhibitory effect on circulating forms of *T. cruzi* similar to that observed previously with ergosterol peroxide [113], but at much lower doses of the compound and is not toxic to mammalian cells [112].

Amphotericin B, used to treat systemic fungal diseases and visceral leishmaniasis, has also been tested for potential trypanocidal activity. This molecule acts by binding to ergosterol in the parasite’s membrane and destabilizing it, which can lead to cell death [114]. Currently, the drug is found in liposome form (AmBisome^®^), and studies point to its efficacy both in vitro and in vivo. Cencig et al. (2011) evaluated the effect of AmBisome treatment on *T. cruzi* parasite load in the Tulahuen strain and trypomastigote form in mice under different therapeutic regimens. Early treatment (1 day after infection) generated the best results, reducing parasitemia as well as parasite accumulations in the heart, liver, spleen, skeletal muscle, and adipose tissue (quantification by RT-qPCR), both in the acute and chronic phases. However, there was an increase in parasite loads when the animals were subjected to cyclophosphamide-induced immunosuppression, indicating that the treatment did not result in a cure [115].

Amphotericin B was also formulated in polyaggregates, albumin microspheres, and sodium deoxycholate micelles and showed good results in in vitro and in vivo assays. The polyaggregates showed IC50 of 0.55 µg/mL and 10.6 µg/mL for epimastigotes and amastigotes, respectively. Cytotoxicity was also evaluated in NCTC929 fibroblasts (CC50 = 96.5 µg/mL). The microspheres showed an IC50 of 0.47 µg/mL and 7.09 µg/mL for epimastigotes and amastigotes, respectively, with CC50 of 111.1 µg/mL. Micelles, on the other hand, showed IC50 of 0.79 µg/mL and 0.07 µg/mL for epimastigotes and amastigotes, respectively, with a CC50 of 221.5 µg/mL, which showed a selectivity index of 3164 for amastigotes. Comparatively, all formulations obtained better results than BNZ and NFX in the tests performed. In the in vivo assay, the micelles showed high toxicity when administered parenterally, but did not show this profile by the oral route, and at the dose of 10 mg/kg, they were able to decrease parasitemia by about 80%. The microspheres also showed parenteral toxicity and were only effective in reducing parasitemia up to 7 days post-infection. However, the formulations were successful in increasing the survival rate [116].

### 4.2. Free Radical Formation

#### 4.2.1. Benznidazole

Cytotoxic metabolites induced by BNZ are the most explored and only alternative currently used for the treatment of CD. It is a nitroheterocyclic that acts through nitrogenous free radicals produced by human nitroreductases that induce covalent modifications of macromolecules: nuclear and mitochondrial DNA, lipids, and proteins. These free radicals can damage the parasite’s DNA and exert an inhibitory effect on protein synthesis and ribonucleic acid synthesis in *T. cruzi* cells [103,117], eventually leading to double-strand breaks in the parasite’s genome [118]. Another mechanism of action of BNZ is through the increase in phagocytosis acting in the elevation of the production of the cytokine gamma-interferon (INF-γ), causing cell lysis [119].

Some studies have been performed using nanoemulsions as carrier systems of molecules with trypanocidal activity. Streck and colleagues (2019) prepared medium-chain triglyceride nanoemulsions by the low energy method and incorporated the drug BNZ, obtaining droplets between 73 and 241 nm and uniform distribution, with PDI below 0.4. The anti-*T. cruzi* activity was evaluated in epimastigotes and trypomastigotes forms of *T. cruzi* strain Y. For the epimastigotes, it was observed that in 24, 48, and 72 h the tested formulations were more effective than the free drug in the same concentration. After 72 h of incubation, the formulations demonstrated IC50 up to 48 times lower than the free drug (0.09 μg/mL for NE and 4.3 μg/mL for free BNZ). This can be justified by the cellular uptake of these drug-laden oily core droplets by the mechanism of endocytosis. For trypomastigote forms, there was no inhibition of parasite growth at the tested concentration of BNZ (50 μg/mL) [120].

One of the main tissue types colonized by *T. cruzi* is the liver tissue, however, once intestinal absorption occurs, BNZ binds to plasma proteins, red blood cells and distributes itself in many tissues not having a selective targeting to the liver [121]. High doses of BNZ are required to achieve therapeutic blood levels against circulating forms. Morilla et al. (2004) used multilamellar liposomes (MLV) loaded with BNZ to evaluate the in vivo efficacy of these systems in delivering BNZ [122]. The tests with the most promising results were intravenous administration in mice infected with trypomastigotes of the pantropic/reticulotropic RA strain, which is efficiently internalized by macrophages [123]. The result demonstrated a threefold greater accumulation in the liver of the mice, and 30% lower concentrations of BNZ in the blood (1.1 g/mL), compared to treatment with free BNZ. However, the relationship between increased selectivity for infected tissue and the therapeutic effect was not direct, with no effect on mouse parasitemia levels.

Another type of nanosystem studied was mesoporous silica nanoparticles (MSNs), which have an ordered pore network structure with hundreds of empty channels offering high surface area. For application in trypanocidal treatment, MSN functionalized with chitosan was studied to allow anchoring of BNZ. An in vitro assay was performed using epimastigotes of the CL Brener clone. It was observed that both pure BNZ and encapsulated BNZ reduced the survival rate of the parasites, however, at a 30-fold lower concentration [56]. Thus, although functionalized MSNs show efficiency in delivering BNZ, more robust in vitro and in vivo studies are needed to explore their trypanocidal potential.

In the study by Tessarolo et al. (2018), inorganic nanoparticles were also used to investigate the effect of calcium carbonate nanoparticles loaded with BZN against epimastigote, trypomastigote, and amastigote forms of *T. cruzi* strain Y. The particle produced showed a size in the range of 27–64 nm by atomic force microscopy. In parasite activity assays on epimastigotes, BZN-CaCO_3_ showed a better trypanocidal effect than free BZN with a sevenfold lower IC50. In tests with trypomastigotes, the nanotechnology system also performed better than the free drug (BZN-CaCO3 LC50 was 1.77 ± 0.58 µg/mL, while BZN LC50 was 66.9 ± 20.3 µg/mL). Finally, in intracellular amastigote forms BZN-CaCO_3_ was able to reduce the percentage of infected cells and the number of amastigotes per cell, consequently reducing the parasite survival rate, which can be justified by the increased drug permeability promoted by nanoparticles [124]. Therefore, a promising BNZ delivery system was developed and its trypanocidal potential in vivo should be explored since it promoted lower toxicity and higher selectivity to *T. cruzi* than the free drug.

Solid dispersion (SD) was a strategy used by Palmeiro-Rodán et al. (2014) to improve the bioavailability of BNZ. Among the carriers tested to obtain the dispersion, better results were observed with a low substitution hydroxypropyl cellulose (L-HPC) ratio 1:3 *w*/*w* drug/carrier-a water-soluble non-ionic polymer. This polymer facilitated the wetting of BNZ and increased the dissolution rate to (85% in 5 min) compared to the pure drug (23% in 5 min). In vivo assays using Y-strain trypomastigotes evaluated that the same SD-1:3 L-HPC formulation showed greater parasite suppression at a dose of 25 mg/kg/day for five days. A total of 60% suppression versus 33% suppression exerted by BNZ was reported [53].

Another group of researchers produced a liquid SD-based formulation using poloxamer 407 [125]. The performance of the system was evaluated in mouse models infected with trypomastigotes of the Tulahuén strain during the acute and chronic periods. In addition to monitoring direct parasitemia, total anti-*T. cruzi* antibodies and parasite load in tissues were measured at 4 or 6 months post-treatment. It was found at the end of the research that the trypanocidal efficacy of SD (60 mg/kg/day) was equivalent to commercial BNZ (50 mg/kg/day) but without manifest side effects or hepatotoxicity.

Mazzeti and colleagues (2020) developed an SEDDS formulation containing BNZ for pediatric use and obtained droplet sizes of approximately 500 nm when incorporating the drug. Tests were performed to evaluate cytotoxicity in uninfected H9c2, HepG2, and Caco-2 cells, trypanocidal action against strain Y trypomastigotes in vitro in H9c2 cells, and activity on parasitemia in an in vivo model in Swiss mice. The formulation was shown to be safe at concentrations below 25 µM, which is almost three times higher than the IC90 of free BNZ. The anti-*T. cruzi* activity tests were performed at the concentrations considered safe and showed a dose-dependent profile on parasitemia growth inhibition, but there was no difference in activity with respect to the free drug. In vivo testing also showed similar results regarding the efficacy and safety of SEDDS compared to free BNZ, where both treatments showed a 57% (4/7) cure rate and ensured 100% survival of the tested animals [12]. Nevertheless, BZ-SEDDS is a promising and innovative pharmaceutical form, especially for pediatric administration.

Scalise and co-workers (2016) prepared polymeric nanoparticles of poloxamer 188 containing BNZ by nanoprecipitation technique [126]. Systems with a size of 63.3 ± 2.82 nm and uniform distribution (PdI < 0.4) were obtained. In the evaluation of cell viability, no morphological change or destabilization of the cell membrane was found. Moreover, after performing the hemolytic assay, the safety of the developed system was observed for not causing lysis in erythrocytes. In vitro and in vivo assays used *T. cruzi* Nicaragua (TcN) strain trypomastigotes. The antitrypanosome activity demonstrated that for 50% of trypanosomes to be lysed (LC50), 49 µg/mL of BNZ and 36µg/mL of NP-BNZ were necessary, a significant reduction. In amastigote assays in VERO cells and cardiomyocytes, it was reported that 25 µg/mL of NP-BNZ achieved the same inhibitory effect as 50 µg/mL of free BNZ, a reduction of half the dose to obtain the same trypanocidal effect. In the acute phase assay in C3H/HeN mice a dose-dependent antiparasitic effect was seen, so animals treated with NP-BNZ (50 and 25 mg/kg/day) survived for at least 50 days, while the infected group treated with NP-BNZ (10 mg/kg/day) exhibited a 70% survival rate after 38 days.

The continuation of this research revealed histopathological data on the inflammation of cardiac tissue produced by the TcN strain [127]. The result showed that infected mice treated with BNZ (50 mg/kg/day) exhibited inflammatory damage similar to that of the untreated infected group. In contrast, a significant decrease in inflammatory cells in cardiac tissue was observed after treatment with NP-BNZ at dosages of 25 and 50 mg/kg/day, with low levels of *T. cruzi*-specific antibodies, and about 40% of animals showed negative CRP results even after immunosuppression. Mice treated with NP-BNZ survived until euthanasia (92 days after infection, dpi). Furthermore, NP-BNZ-treated Vero cells led to a significant increase in reactive oxygen species (ROS) production, suggesting that the drug incorporated into the nano-system follows the same pathway involving nitroreductase II as free BNZ.

The influence of the therapeutic scheme of these nanoparticles with BNZ in lower doses, in the chronic phase of *T. cruzi* infection (Nicaragua in the trypomastigote form) in C57BL/6J mice was also evaluated [128]. At 90 days after infection, animals were treated with NP-BNZ in two therapeutic regimens: (i) doses of 25 or 50 mg/kg every 24 h; (ii) doses of 50 or 75 mg/kg every 7 days for 91 days (totaling 13 applications). The intermittent administration of 75 mg/kg of BNZ-NP was as effective as the intermittent administration of 100 mg/kg of free BNZ, which corresponds to a 25% dose reduction to obtain the same effect. Both regimens showed good results regarding parasitemia elimination, reduction of *T. cruzi*-specific antibodies, and INF-γ-producing cells. In addition, improvement was observed in electrocardiographic parameters and in signs of cardiac inflammation and fibrosis compared to the untreated groups. However, the total dose at the end of treatment is substantially higher in the continuous regimen, which reveals that the intermittent regimen may be safer, as it uses smaller amounts of BNZ, and may allow longer periods of treatment with fewer undesirable effects, which may be effective against latent forms of the parasite.

García et al. formulated an oral multiparticulate drug delivery system (MDDS). This was developed based on interpolielectrolyte complexes obtained by the ionic interaction of two oppositely charged Eudragit^®^ polymethacrylates to transport BNZ. In the study, the combination of polymethacrylates allowed for an interesting multi-kinetic in vitro release profile to reduce the side effects of the drug. In the in vivo assay using *T. cruzi* trypomastigotes of Tulahuen strain in the acute phase, it was observed that doses of 100 mg/kg/day of free or MDDS-loaded BNZ were sufficient to nullify parasitemia. Notably, MDDS-BNZ reduced levels of liver injury markers compared to free BNZ [129]. Continuation of this research revealed that intermittent treatment administered every 5 days at a dose of 100 mg/kg/day (13 doses total) of BNZ loaded in MDDS induced the lowest cardiac parasite burden, indicating improved efficacy with lower total BNZ dose [130]. This investigation was important in verifying that sustaining the plasma concentration of the drug for longer lapses of time avoids high peaks in plasma levels and helps to reduce the toxicity of BNZ.

#### 4.2.2. Nifurtimox

NFX is a prodrug, which is activated by the enzyme NADPH-cytochrome P-450 reductase that reduces its nitro group (NO2) to NO2U- and undergoes an oxireduction cycle with oxygen that leads to the formation of superoxide radicals (O2U-) and return of NFX to the prodrug form. Superoxide undergoes dismutation into molecular oxygen and hydrogen peroxide (H_2_O_2_). O_2_U- and H_2_O_2_, in a medium containing Fe_3_+ produce hydroxyl free radicals (OHU) that interact with macromolecules of the parasite, causing damage. The NO_2_U-radical also blocks the enzymes glutathione and trypanothione reductases that are responsible for controlling intracellular oxidative stress [103,131].

The first nanotechnology system reported in the literature used for the treatment of CD was produced by Gonzalez-Martin and collaborators (1998) [131], where polyethylcyanoacrylate nanoparticles were produced for the incorporation of NFX. The incorporation efficiency was 33.4 ± 2.1%. The release study showed that at pH 7.4, 65% of NFX was released during a 6 h period. Only in vitro assays were performed using the epimastigote form of a clone obtained from *T. cruzi* isolated from a patient with the chronic form of the disease in Northern Chile. In the anti-epimastigote study, Nano-NFX at the concentration of 0.2 µg/mL showed 98.9% inhibition percentage (IC50 = 0.015 ± 0.006 µg/mL) while free NFX was 40% (IC50 = 0.683 ± 0.269 µg/mL). For trypanocidal activity in amastigotes, the results showed that Nano-NFX showed a higher percentage of inhibition compared to the standard NFX solution at all concentrations tested, especially at the concentration of 0.1% which inhibited 97.8% against 54.6% of the free drug after 72 h in the same concentration. When analyzed by electron microscopy, morphological changes in the parasite were reported, such as accumulation of nuclear chromatin, and lysis of the parasite membrane, and no changes were found in the host cell.

#### 4.2.3. Photodynamic Therapy

Hypericin is a polycyclic aromatic quinone that is extracted from the plant *Hypericum perforatum* and used for photodynamic treatment, which has recently been studied for anti-*T. cruzi* activity. Photosensitizing molecules, when in contact with light and oxygen produce reactive oxygen species and singlet oxygen which are responsible for causing cellular damage [132]. However, hypericin has high molecular weight and high lipophilicity, which hinders its solubility in biological fluids and compromises its arrival at its site of action [133].

Morais and colleagues (2019), tested the efficacy of hypericin against *T. cruzi* (strain Y) trypomastigotes in three different nanocarriers, polymeric micelles of Pruronic™ F-127 and P-123 and phospholipid liposome 2-dipalmitoyl-sn-glycero-3-phosphocholine (DPPC), both in the presence and absence of light. The polymeric micelles showed efficacy at concentrations higher than 0.8 µM, and EC50 around 0.3–0.4 µM for experiments in the light and 6–8 µM for experiments conducted in the dark. The liposome showed results of 2.22 ± 0.52 µM and 10.84 ± 0.47 µM for light and dark, respectively. A formulation with free hypericin was also tested and obtained similar results to micelles in the lighted assay (0.37 ± 0.08 µM), however, in the dark the EC50 was 14.84 ± 0.61 µM [134].

### 4.3. Purine Salvage Inhibitor

Allopurinol is a hypoxanthine analog that acts in the purine rescue pathway as an alternative substrate of the hypoxanthine-guanine phosphoribosyltransferase enzyme of T. cruzi, forming the nucleotide 4-aminopyrazolpyrimidine triphosphate, which is incorporated into RNA, destabilizing the strand, and interfering with protein and new purine synthesis [135,136]. However, the doses required for a good trypanocidal activity are high and bring important adverse effects, arising the need for mechanisms capable of potentiating their effect.

Thus, Gonzalez-Martin et al. (2000), used poly(ethyl)cyanoacrylate polymeric nanoparticles also to encapsulate Allopurinol. The incorporation efficiency was 69 ± 3.4 µg/mg of the nanoparticle. In the release assay, the drug was shown to have a pH-dependent release, at pH 7.4 it released 34.3 ± 0.4 µg/mL after 6 h of incubation while at pH 1.3, it was only 14.4 ± 0.8 µg/mL. The in vitro assay occurred using an epimastigote culture of a clone (CA-1) obtained from *T. cruzi* isolated from a patient with the chronic form of the disease in Northern Chile. The highest trypanocidal activity occurred at the concentration of 16.7 µg/mL, with 91.5% (IC50 = 0.5 ± 0.1 µg/mL) using the nanoparticle system with allopurinol, while at the same concentration, the allopurinol standard solution showed only 45.9% activity (IC50 = 37.3 ± 5.0 µg/mL). It is worth noting that allopurinol presents less toxicity than NFX, and is thus more strategic in the construction of nanocarriers with this drug for a potential treatment for CD [137,138,139].

### 4.4. Enzyme Inhibition in T. cruzi

#### 4.4.1. Inhibition of Cruzipain

Cruzipain is the most abundant protein in *T. cruzi*, which belongs to the proteases or peptide hydrolases family. Is an important virulence factor of the parasite involved in several crucial steps in the interaction with mammalian cells [140,141] and is present on the surface and in the flagellar sac of all three evolutionary forms, and also plays an important biological role in the parasite in cell remodeling during the transformation from epimastigote to the infective metacyclic stage [142,143].

Nitric oxide (NO) plays an important role in defense against pathogens. During *T. cruzi* infection, pro-inflammatory signals are produced through the production of IFN- γ, IL-12, and TNF-α that generate substantial intracellular NO production, which is essential for antiparasitic activity [144,145,146]. Studies indicate that NO is the mechanism by which IFN-γ controls *T. cruzi* infection by noting that it blocks the parasite’s life cycle in vitro and in vivo [147]. Thus, the antichagasic action of NO depends on factors such as the induction of NO-synthase (NOS2), generating NO in response to infection that can inhibit the catalytic activity of cruzipain.

Another important factor in trypanocidal action is the oxidative stress generated by phagocytes (NO and O_2_^−^), where NO reacts with the superoxide radical (O_2_^−^) to generate peroxynitrite (OONO−), a potent oxidant and nitrating molecule [148,149,150]. On the other hand, the parasite produces an inflammatory response that leads to evasion of the host’s immune response, and among the mechanisms is the induction of apoptosis in various cells and immunosuppression of the host. In both cases, NO production is compromised, because the levels of pro-inflammatory cytokines such as TGF-β and PGE2 are elevated. In order to increase the levels and lifetime of NO, some studies have been conducted using nanotechnology to deliver exogenous NO donors.

An interesting alternative in the fight against CD, was tested by Seabra et al. (2015) by administering exogenous NO donors encapsulated in chitosan/sodium tripolyphosphate (TPP) polymeric nanoparticles. Mercaptosuccinium acid (MSA) containing thiol was nitrosated with sodium nitrite (NaNO_2_), leading to the formation of S-nitroso-MSA (spontaneous NO donor). They obtained nanoparticles of 270 to 500 nm with an incorporation efficiency of 99%. In in vitro assays, the antitrypanosomal activity using strain Y was dose-dependent for both epimastigotes and trypomastigotes, decreasing the number of parasites after 24 h with IC50 = 252 µg/mL. In the in vivo assay, murine was infected intraperitoneally with trypomastigote forms to obtain macrophages with amastigotes. The results showed that the concentration of 200 µg/mL showed activity against amastigotes. Further studies from the same group reported morphological and biochemical changes induced by these NO-releasing nanoparticles, such as cell shrinkage, cell cycle arrest, depolarization of the mitochondrial membrane, and exposure of phosphatidylserine on the cell surface, indicating that epimastigote death is associated with the apoptotic pathway [151,152]. Therefore, it has proven to be an interesting approach against CD being necessary for the initiation of in vivo research for more knowledge about its trypanocidal potential.

Dithiocarbazates exhibit remarkable biological and pharmacological properties, including trypanocidal activity by S-dithiocarbazate derivatives. Among the derivatives, 5-hydroxy-3-methyl-5-phenyl-pyrazoline-1-(S-benzyl dithiocarbazate) (H2bdtc) exhibited significant trypanocidal activity [14]. A study using the Tulahuen strain in vitro in epimastigotes revealed a high antiproliferative activity with IC50 = 15 µM compared to IC50 = 1800 µM of the standard drug, BNZ. In the values with amastigotes, H2bdtc stood out with IC50 = 0.6 µM compared to IC50 = 2.5 µM by BNZ, revealing a potential compound for the treatment of CD. Its mechanism of action is not yet fully elucidated, but due to the pyrazole and dithiocarbazate portions of the molecule, which are similar to triazoles and thiosemicarbazones, an inhibition of cruzaine is suggested [153]. However, the lipophilic character of H2bdtc limits its administration and results in low oral bioavailability, thus new delivery systems were developed to circumvent this limitation.

Lipids have excellent physiological acceptability and are therefore extremely strategic in the construction of solid lipid nanoparticles (NLS) [154]. NLS consisting of sodium taurodeoxycholate, stearic acid, and soy lecithin were produced by microemulsion for incorporation of H2bdtc and showed sizes in the 130 nm range with homogeneous distribution (PDI < 0.3). In vitro assays in Y-strain trypomastigotes revealed that both free H2bdtc and H2bdtc-SLNs exhibited trypanocidal activity similar to BZN. But in vivo results reported that H2bdtc-SLNs eliminated 70% of circulating parasites at peak infection, while free H2bdtc and the positive control BZN eliminated 48 and 15% of parasites, respectively. When analyzing the cardiac and liver tissues of the surviving animals, they observed that infected mice treated with H2bdtc-SLNs showed reduced cardiac inflammation and cardiac lesions were absent. In addition to reducing the liver damage caused by the parasite, they decreased inflammatory infiltration in the liver and liver toxicity more effectively compared to the free form and the standard drug, BNZ. Thus, allowing the conclusion that H2bdtc is a potent trypanocidal agent [155].

#### 4.4.2. Inhibition of Carbonic Anhydrase

*T. cruzi* also encodes α-carbonic anhydrase (CA, EC 4.2.1.1) a metalloenzyme with high catalytic activity (TcCA) [156]. CAs catalyze a simple physiological reaction, the reversible conversion of CO_2_ to bicarbonate ions and protons. The active site of α-ACs contains a zinc ion (Zn^2+^), which is essential for catalysis [157,158]. This enzyme is susceptible to inhibition by sulfonamides that incorporate the R-SO2NH_2_ portion by binding to the Zn^2+^ ion of the enzyme. This generates impairment in the formation of the nucleophilic zinc hydroxide species of the enzyme, which is responsible for catalyzing the hydration of CO2 into bicarbonate [159,160]. However, the best TcCA inhibitors detected in the study by Pan et al. (2013) despite showing in vitro inhibition constants in the range of 61.6–93.6 nM parasitemia, did not obtain any in vivo effect on pathogen growth [158], which may be related to the fact that sulfonamides are poorly penetrating agents across biological membranes, a phenomenon already found for the inhibition of CAs from nematodes or fungi [161,162,163,164].

A single study, to date, has been found in the literature with nanotechnology aimed at improving the delivery of sulfonamide derivatives against *T. cruzi*. In order to increase their bioavailability and permeability across membranes, nanoemulsions were formulated in clove oil (*Eugenia caryophyllus*). Sulfonamides 3F, 3G, 3W, 5B, 5C, and 5D studied previously by the same group were the drugs tested [165]. To study the anti-*T. cruzi* effects, two different strains (Dm28c and Y) of *T. cruzi* epimastigotes were selected. The IC50 values of the sulfonamide NEs were lower than that of BNZ (20.63 µM) for the epimastigote forms of both strains. Sulfonamide 3F showed the best activity with IC50 of 3.54 µM. However, all derivatives showed cellular toxicity against RAW 267.4 macrophage cells. Flow cytometry analyses evidenced that NEs containing the sulfonamides 3G, 5D, and 3F led to cell death by necrosis in the following proportions of 82.41%, 81.26%, and 57.03%, respectively, for the Dm28c strain, being more effective than the reference drug BNZ (effect of 51.16%) [166]. These effects can be justified by the high ability to permeate membranes presented by nanoemulsions, thus interfering in the life cycle of the pathogen. Nevertheless, in vivo tests are still needed to observe the maintenance of the anti-*T. cruzi* activity.

#### 4.4.3. Inhibition of Trypanothione

Lycnofolide (LYC) is a sesquiterpene lactone of natural origin, isolated from *Lychnophora trichocarpha,* and exhibits anti-*T. cruzi* activity. Although its mechanism is not well elucidated, it is believed that it acts by covalently binding to the trypanothione enzyme in the parasite, which has a function analogous to glutathione for mammalian cells, inactivating it and leading to increased intracellular oxidative stress and is essential and exclusive in trypanosomatids [167,168]. However, due to its physicochemical characteristics not favoring its oral bioavailability, its potential was studied using two types of nanosystems. One composed of poly-ε-caprolactone (PCL) with encapsulation efficiency greater than 95%, and particle size of 182.5 ± 3.2 nm, and another of poly(lactic acid)-polyethylene glycol (PEG-PLA) with 100% incorporation efficiency and size of 105.3 ± 2.3 nm [169]. For the studies in mice infected with the trypomastigote form of the CL or Y strains a treatment with 10 and 20 doses (2 mg/kg/day) was used. It was observed that 100% cure occurred in animals infected with strain Y treated with LYC-PLA-PEG-NC for 20 days (treatment started 7 days after infection), while those treated with BNZ and LYC-PCL-NC obtained less than 75% and 62.5% cure, respectively. The result of better LYC-PLA-PEG-NC performance can be explained because the PEG chains make the particles stealthier, delaying the rapid removal of the system from the bloodstream by macrophages, and increasing the exposure time of the trypomastigotes to the drug.

### 4.5. Induction of Autophagy

Autophagy plays a protective role in the mammalian against *T. cruzi* infection [170]. In the study by Casassa et al., 2019, mice deficient in Beclin-1 (a protein essential for autophagosome biogenesis and maturation) showed higher parasitemia, heart parasite nests, and mortality rates compared to the control group [171,172].

Ursolic acid (UA), a pentacyclic triterpene of natural origin found in different plant species, is a compound with several biological activities that has its anti-*T. cruzi* activity attributed to the ability to increase autophagosome formation, inducing autophagy in both macrophages and H9c2 cardiac cells [170]. Oral administration of 50 mg/kg AU reduced parasitemia peaks in mice infected with *T. cruzi* strain Y in an acute infection model [173]. However, its bioavailability is impaired by its very low water solubility, which reinforces the need for systems that improve biopharmaceutical aspects.

Solid dispersions (SD) with Gelucire 50/13 were generated in order to improve the dissolution profile of the lipophilic UA molecule. The trypanocidal activity was performed using Y-strain trypomastigotes. The results obtained demonstrated that the 250 μM concentration of AU caused the greatest lysis of *T. cruzi* strain Y (48.27 ± 4.20), with SD showing greater trypanocidal activity than the physical mixture, exhibiting an IC50 of 219.2 and 396.7 μM, respectively [13]. Despite the higher dissolution of AU in the SD containing 95% Gelucire and silicon dioxide, in vivo tests are still needed to determine the effects of AU carried by the SD in the treatment of CD.

Seeking to improve the bioavailability of the molecule, poly-ε-caprolactone (PCL) polymeric nanoparticles were also developed via the nanoprecipitation technique [174]. Nanoparticles with a size of 173.2 ± 7.28 nm were obtained with no evidence of cytotoxicity in fibroblast cells (LLC-MK2), whereas it showed 56.85 ± 5.70% inhibition against trypomastigotes (Y strain) at the concentration of 30 μM during 24 h of incubation. In in vivo studies performed in male C57BL/6 mice, it was observed that NP-UA reduced parasitemia by up to 3.5-fold when compared to the BNZ-treated group. Compared to the parasite inhibition results of Eloy et al. (2012), it is noteworthy that with an eight-fold lower dose, similar inhibition percentages were obtained [13].

Another study performed by Oliveira et al. (2017) evaluated the activity of AU-loaded Capryol^®^ 90 oil-based nanoemulsions with droplet sizes of approximately 37 nm and unimodal distribution (PdI < 0.2). The trypanocidal activity for NE of AU and free BNZ was evaluated in vitro on amastigotes forms of CL Brener clone strain B5. The results indicated that the IC50 of NE was 18 μM while that of BNZ was 4.1 μM, however the CC50 of NE was 396.4 μM, demonstrating lower toxicity when compared to free BNZ, which showed CC50 of 165 μM, showing that nanoemulsified AU was safer than the standard drug for the treatment of CD [175].

### 4.6. Inhibition of Tubulin Assembly

(−)-hinocinin (HNK) belongs to the dibenzylbutyrolactone class of lignan compounds and has been investigated for exhibiting various biological activities, including trypanocidal activity [176], as demonstrated by Souza et al. (2005) who evaluated the anti-*T. cruzi* activity of HNK in vitro against free amastigote forms of *T. cruzi* strain Y. The results showed an IC50 0.7 μM compared to BZN IC50 0.8 μM. In view of its trypanocidal activity, Saraiva and collaborators (2007) tested hinocinin against the epimastigote and amastigote forms of the CL clone of *T. cruzi*. For the epimastigote form, the compound showed high trypanocidal potential, with IC50 0.67 μM, while BNZ showed IC50 of 30.89 in the study. For the amastigote form, the inhibitory activity was considered low. In contrast, it was observed in in vivo assays that treatment with HNK promoted 70.8% parasitemia reduction at the parasitemic peak in the trypomastigote form of strain Y, while BNZ showed approximately 29.0% parasite reduction [177].

However, despite the promising results, its mechanism of action on the parasite is not yet elucidated. A mechanism of action similar to (−)-cubebin is suggested by structural similarity, a compound from which HNK is obtained by partial synthesis. The mechanism of (−)-cubebin occurs in the tubulin assembly inhibition pathway, molecular docking simulations indicated a hydrophobic interaction with α-tubulin residues (Ser178 and Ala180) and hydrogen bonds with β-tubulin residues [178]. The study helped to understand the trypanocidal activity of this lignan. Since the basis for parasite cytoskeleton architecture is basically organized and stabilized by stable microtubules, which in turn are composed of heterodimers of α/β-tubulin. Thus, inhibition of the assembly of these proteins can lead to disruption of the cytoskeleton and parasite death [179].

A microparticle system was the only one used so far as a strategy for HNK drug delivery. The objective was to promote sustained release and to test the trypanocidal effect in mice infected with *T. cruzi* trypomastigotes (clone CLB5). Poly(D,L-lactide-co-glycolic acid) (PLGA) microparticles loaded with HNK had an average diameter of 0.862 µm. Treatment of infected mice with 40 mg/kg HNK-loaded microparticles every other day was able to promote a significant decrease in parasitemia levels compared to those recorded in untreated controls. Furthermore, administration of HNK-loaded microparticles was able to reduce the number of parasites more than treatment with 20 mg/kg/day HNK not only at the parasitemic peak but also over the course of infection. Therefore, the vehicle to deliver HNK may improve trypanocidal activity, but further study in other strains and amastigote forms is needed [180].

### 4.7. Immunomodulation

Imiquimod (IMQ) is an imidazolquinoline that has an immunomodulatory capacity by increasing the release of cytokines, consequently increasing the activity of dendritic cells and macrophages via activation of Toll-like receptor 7 [181] and therefore, has the ability to induce a greater response of the immune system on the parasite. This molecule was incorporated into nanoarcheosomes, which are nanocarriers composed of lipids extracted from the archaebacterium *Halorubrum tebenquichense*. This system was tested for its ability to induce protection against *T. cruzi* infection, and survival guarantees were verified, where 100% of the animals treated with the nanocarrier IMQ (nanoarc-IMQ) survived until the end of the experiment, in contrast to the animals treated with free IMQ, which died on the 33rd day post-infection. The decrease in parasitemia was evaluated according to the area under the curve (ASC) of a graph of parasitemia × days post-infection, where nanoarq-IMQ showed an ASC of 213.6 ± 49.7, significantly lower than the negative control group (PBS), the blank system and free IMQ, 528.5 ± 78.6, 426.9 ± 104.9 and 494.9 ± 89.0, respectively. The positive control was made with BNZ, and its ASC was 0. In the research for specific antibodies, animals treated with nanoarq-IMQ presented four times more IgG1 and IgG2a than animals treated with BNZ, reinforcing the immunomodulatory capacity of IMQ. The tested system was also able to significantly decrease the inflammatory response in cardiac and skeletal muscles [182].

### 4.8. Inhibition of RNA Synthesis

Actinomycin D (ActD) is an antibiotic obtained from a filamentous bacterium, *Streptomyces*. Its molecules bind non-covalently to DNA, preventing the opening of the double strand and consequently inhibiting RNA transcription and synthesis [183]. Soon when used in *T. cruzi* trypomastigotes, a loss of parasite multiplication ability was observed at different ActD concentrations (10, 20, and 50 µg/mL) [184].

Rigid lipid nanosomes in the form of hollow pentagonal dodecahedrons (ISCOMs) were produced for the encapsulation of ActD [185]. The ISCOM was functionalized with compounds containing vinyl sulfone lipids capable of binding functional IgGs on their surface, favoring the recognition and transport of these nanocapsules to specific *T. cruzi* cells. The results showed that a dose of 50 ng/mL of ActD administered in free form was able to kill 100% of epimastigotes of the Maracay strain in 24 h, while the concentration of the antibiotic encapsulated in functionalized ISCOMs generated similar efficacy with a 196.8-fold lower dose (25.47 × 10^−2^ ng/mL), a very expressive dose reduction. Further studies in other parasitic forms and in vivo to observe the ActD anti-*T. cruzi* potentials are needed.

### 4.9. Other Mechanisms

#### 4.9.1. N,N′-Squaramide 17

N,NAs′-squaramides are amide-type compounds that have both electron donor and electron acceptor groups and are capable of multiple interactions with complementary sites. The study produced by Olmo et al. (2014) verified the potential of squaramide-based compounds with antichagasic activity. The results suggested squaramide 17 as the ideal structure among the tested compounds. The antichagasic activity results of compound 17 showed higher activity than in the reference drug BNZ (IC50 values were 9.4 ± 0.4 and 8.5 ± 0.4 μM in epimastigotes and amastigotes forms, respectively, compared to 15.9 ± 1.1 and 23.3 ± 4.6 μM for BNZ). It also found a decrease in infection rate for compound 17 of 67% compared to BNZ 20% when analyzing parasite propagation in Vero cells for 10 days [186].

Furthermore, in vivo studies have shown that squaramide 17 (25 mg/kg) was the most effective, causing a 67% reduction in the number of trypomastigotes at the end of the experiment on day 40 compared to the untreated control [187]. The mechanism of action of this compound is not well elucidated, but it is known that in the lysosome hydrolysis of squaramide occurs, separating it into squaric acid and two amides. Thus, squaric acid can interact with some molecules essential for the intracellular metabolism of the parasite, generates modifications in the glycolysis cycle of *T. cruzi*, and produces cytoplasmic and mitochondrial alterations causing a reduction in its replication inside the cell [188].

Liposome mimetics of lipids extracted from RAW 264.7 cells were produced to potentiate the activity of N,N′-squaramide 17 (MLS) [189]. In the in vitro study, MLS induced trypanocidal activity in epimastigotes and amastigotes forms of *T. cruzi* (IC50 = 15.85 ± 4.82 µM and 24.92 ± 4.80 µM). Although BNZ showed lower IC50 on amastigotes (IC50 = 15.85 ± 4.82 µM and 24.92 ± 4.80 µM), MLS showed a twofold increase in efficacy on the amastigote form compared to N,N′-squaramide 17 (IC50 = 51.18 ± 4.91). It was also observed that both free drugs (S) and MLS showed less cytotoxic activity in VERO cells, with CC50 of 736.21 μM and 1199.50 μM, respectively, compared to BNZ (CC50 = 284.44 ± 1.25 μM). Therefore, MLS showed more selectivity for amastigotes with a selectivity index (SI) value of 48.14, while S was 14.38, demonstrating the benefits of the nanostructured system. By SEM analysis it was seen that MLS was able to induce structural changes in the parasites such as cracks with a diameter of approximately 200 nm in the epimastigote cytoplasm membrane. However, despite the promising results, in vivo studies are needed to confirm the potential of MLS.

#### 4.9.2. Corn Ear Xylan

Xylan is a polymer found in the cell wall of grasses and dicotyledons, formed by D-xylose monomers linked by β14 bonds. Corn cob xylan has been studied for various biological activities, such as antioxidant, antibacterial, and antitumor. In addition to D-xylose, this xylan has O-methyl-D-glucuronic acid and L-arabinose (19:2:7). This compound was incorporated into inorganic silver nanoparticles and its activity was evaluated in vitro on Y-strain epimastigotes. The parasites were incubated with the nanoparticles with xylan (NX) for 24 and 48 h at doses of 2.5, 10, and 100 μg/mL, using BNZ as a positive control, and the activity was verified by the MTT method. NX at the concentration of 100 μg/mL was able to decrease MTT reduction by 82% for 24 h and 95% for 48 h of incubation, which was not observed in free xylan. No significant cytotoxicity was observed in RAW and 3T3 cells at the highest concentration tested (1000 μg/mL). The mechanism of action of xylan is not fully elucidated, but the study found that parasite cell death occurred by necrosis [190].

### 4.10. Discussion

In this review, we investigate and discuss the results obtained in in vitro and in vivo studies regarding the trypanocidal effect of several drugs incorporated into different nanosystems for the treatment of Chagas disease. We grouped the drugs studied according to their cellular target in Trypanosoma cruzi, such as drugs that act on ergosterol, free radical formation, enzyme inhibition (cruzipain, carbonic anhydrase, trypanothione synthetase), induction of autophagy, inhibition of tubulin assembly, immunomodulation, inhibition of RNA synthesis and others, as shown in Table 2.

The data show the influence of the application of nanotechnology on the therapeutic effects of drugs, since, in most studies, the anti-*T. cruzi* effect is greater when the drug is trapped in a nanosystem. Moreover, even in cases where the trypanocidal activity was equivalent to a free or reference drug (benznidazole), there was a reduction in toxicity or even an improvement in solubility.

A limitation of the treatment of *Trypanosoma cruzi* infection is the parasite’s resistance to the commercial drugs approved for use (nifurtimox and benznidazole). However, the research reported in this review does not explore how nanosystems can act on this. Moreover, despite the relevant results, none of these nanotechnology studies applied to benznidazole, nifurtimox, or new drugs have advanced to the clinical trials for the treatment of Chagas disease. All this demonstrates the importance of further studies in this direction and reinforces how much the disease is neglected.

All this research so far contributes to the development of optimal formulations for the treatment of the disease. The more we explore, the more we will know about the interactions between nanosystems and *Trypanosoma cruzi*, and thus we will have potent candidates for anti-Chagasic drug delivery agents, which may be very useful in future clinical trials for Chagas disease.

## 5. Conclusions

Despite different efforts in the search for new compounds with biological activity and their delivery in different delivery systems, the treatment of Chagas disease remains without an effective solution. Many studies have stopped in vitro assays testing mainly the trypomastigote and epimastigote forms. The preclinical data analyzed here have not yet clearly shown the successive elimination of intracellular parasites, and therefore it is not possible to state that the strategies described will be effective for the treatment of the chronic phase of the disease.

Among the challenges to be overcome, the high variability of *T. cruzi* strains and the lack of uniformity and standardization of in vitro/in vivo test models affects the possibility of comparison between studies. Passive targeting of novel delivery systems to the heart in the chronic stage is a key outstanding question. A careful evaluation of toxicity and more effective regimens should be established. Another important point is that the association of drugs that do or do not share the same mechanism of action should be tested in nanosystems, with a view to evaluating the promotion of increased therapeutic efficacy.

Benznidazole and nifurtimox are the only alternatives approved by regulatory agencies to date. In this article, new interventions, insights, and treatment strategies were discussed for both drug delivery systems, to potentiate the action against *Trypanosoma cruzi* infection. Both for the already approved drugs and for new molecules in the test phase it was possible to observe in the studies improvement in cellular uptake, increase in the absorption profile, aqueous solubility, and bioavailability studies, in addition to the reduction of drug toxicity by targeting the parasite. In this context, the use of drug delivery systems seems as an interesting and promising alternative to improve pharmacological responses to Chagas disease.

## Figures and Tables

**Figure 1 ijms-24-13778-f001:**
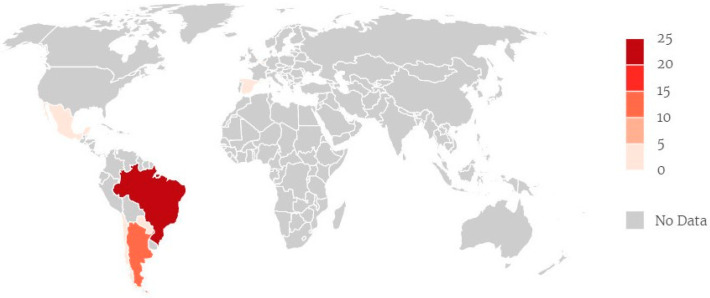
Distribution of included articles (sample) by country. Brazil (21 studies), Argentina (11 studies), Chile (03 studies), Spain (02 studies) and Mexico, Paraguay, Belgium (01 study each). Note: Created with mapinsecond.com.

**Figure 2 ijms-24-13778-f002:**
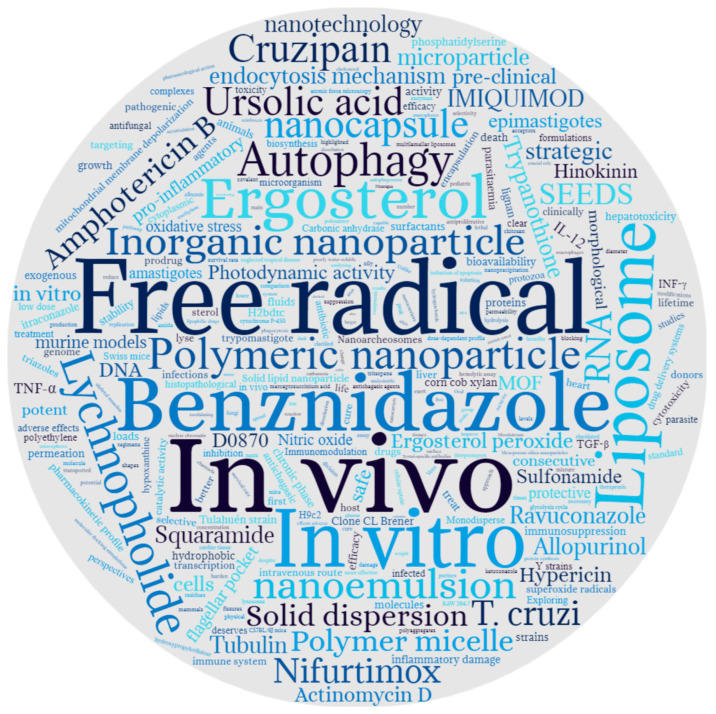
Word cloud created from abstracts of the 40 selected studies. The size of each word is proportional to the number of times it has been cited throughout the text. Note: created with wordclouds.com. Accessed on 18 September 2022.

**Figure 3 ijms-24-13778-f003:**
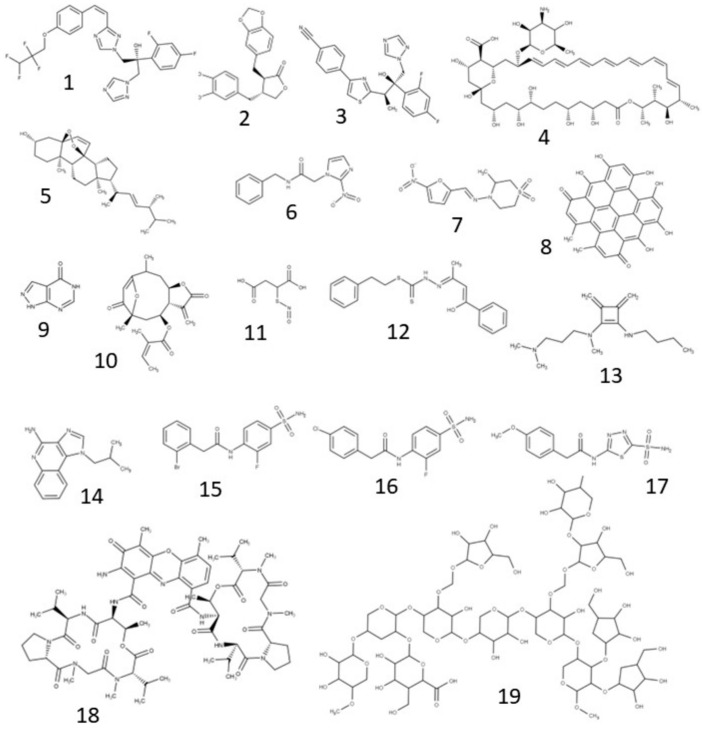
Chemical structure of the major compounds evaluated as antichagasic agents (in vitro and/or in vivo evidence). (**1**) D0870, (**2**) hinokinin, (**3**) ravuconazole, (**4**) amphotericin B, (**5**) ergosterol peroxide, (**6**) benznidazole, (**7**) nifurtimox, (**8**) hypericin, (**9**) allopurinol, (**10**) lychnopholide, (**11**) S-nitroso-MSA, (**12**) H2bdtc, (**13**) N-N’-Squaramide 17, (**14**) imiquimod, (**15**) sulfonamide 3G, (**16**) sulfonamide 3F, (**17**) sulfonamide 5D, (**18**) actinomycin D e, (**19**) corn cob xylan.

**Figure 4 ijms-24-13778-f004:**
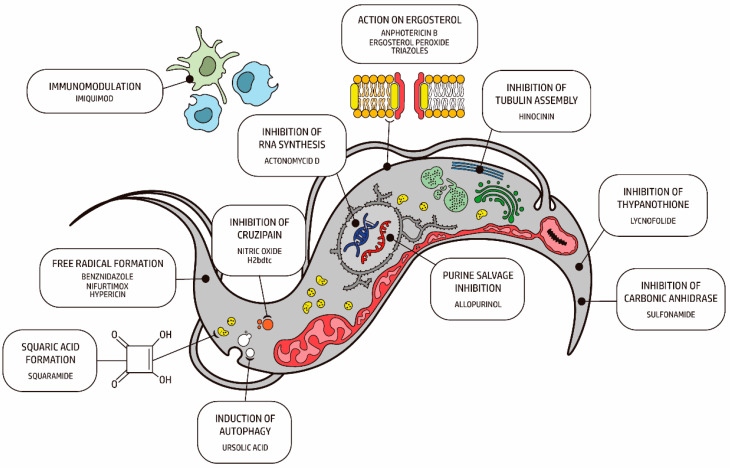
Schematic representation of a trypomastigote of *Trypanosoma cruzi* and the main cellular targets of the investigational compounds in the pre-clinical phase.

**Table 1 ijms-24-13778-t001:** Patent-related nanocarriers for treatment of Chagas disease.

Title of the Patent	Drug/Compound	Nanocarrier	Publication Number	Inventor	Country
Pharmaceutical compositions containing sesquiterpene lactones belonging to the class of furan heliangolides for the treatment of parasitic infections and tumors	Sesquiterpene lactones belonging to the class of furan heliangolides (Lycnofolide)	Polymeric nanoparticles	WO 2013/059898 A1 PI 1106302-5 A2	Mosqueira et al., 2013 and 2012 [19,20]	PCT and BRA
Nanoplatform production process for encapsulation and carrying of drugs and nanoplatform prepared from this	Benznidazole	(BNNS)-Chitosan nanoparticles	BR 102020019797-5 A2	De-Sousa et al., 2020 [21]	BRA
Pharmaceutical compositions of posaconazole and benznidazole with increase in dissolution	Pasoconazol and Benznidazole	Solid dispersion	BR 102016023800-5 A2	Rolim-Neto et al., 2016 [22]	BRA
Pharmaceutical formulations containing benznidazole and mof’s association for technological obtainment of drug delivery systems	Benznidazole	Metal–organic structures	BR 102016003408-6 A2	Rolim-Neto et al., 2016 [23]	BRA
Compositions and process of production of liquid form administration based on emulsion lipid systems, of emulsion, microemulsion and/or nanoemulsion type, containing benznidazole for the treatment of the Chagas disease	Benznidazole	Micro/ nanoemulsions	BR 102012019428-7 A2	De-Oliveira et al., 2012 [24]	BRA
Composition and production method of nanoparticulate systems for modified release of Benznidazole	Benznidazole	Polymeric or lipid nanoparticles	BR 102017024448-2 A2	Da-Silva-Junior et al., 2017 [25]	BRA
Compositions pharmaceutical and veterinary in Micro- and nanostructured forms and forming micro- and nanostructures in the gastrointestinal tract containing benznidazole and its derivatives and their biological applications	Benznidazole	Self-emulsifying	BR 102013023902-0 A2 WO 2015/039199 A1	Mosqueira et al., 2013 and 2015 [26,27]	BRA and PCT

**Table 2 ijms-24-13778-t002:** In vitro and in vivo assessment of nanocarriers developed for Chagas disease treatment.

Nanoformulation	Drug	Cellular Target	Size/PdI/EE/DL	Strain/Evolutionary Stage	Outcomes
Polymeric nanoparticles	DO870, Itraconazole and Ketoconazole	Action on ergosterol	100–200 nm/-/90, 87 and 92%/-	CL Brener and Y/Trypomastigote	The survival of the animals treated with intravenous administration of NP-DO870 at 3 mg/Kg/day is comparable to results observed to the group trated with free benznidazole at 100 mg/Kg/day and induced at 90% and 60% cure rate for strain CL and Y, respectively.
Self-emulsifying delivery System	Ravuzonazole	Action on ergosterol	100–250 nm/>0.4/-/-	Y/Amastigote	RAV-SEDDS increased the inhibition activity against amastigotes ~1.8-fold compared to free RAV at doses equivalent to IC50 (0.1 nM).
Metal–organic frameworks	Ergosterol Peroxide	Action on ergosterol	100 nm/-/-/-	TDIM/MEX/2014/LJ01/*T. cruzi*/trypomastigote and epimastigote	MOFs-EP showed a coparenting effect to free EP for both epimastigote and trypomastigote. However, at lower doses.
Liposome	Amphotecirin B (AmBisome)	Action on ergosterol	-	Tulahuen/trypomastigote	The animals treated with AmBisome promote the prevention of mice from fatal issue in the acute phase and significantly reduces parasite loads in organs and adipose tissue in both phases. However, does not have complete cure.
Polyaggregates, albumin microspheres and micelles	Amphotericin B	Action on ergosterol	30 nm–1–10 µm/-/-/-	CL-B5 and Y/Amastigote, trypomastigote and epimastigote	In vitro studies have shown that in amastigote form the selectivity index was higher to micelles being more than eleven times higher than benzinidazole. In vivo assay have shown high parenteral toxicity on micelles but not orally. At a 10 mg/kg dose, they reduced parasitemia by approximately 80%. Microspheres also displayed parenteral toxicity and were effective in reducing parasitemia up to 7 days post-infection.
Nanoemulsion	Benznidazole	Free radical formation	73.61–241.60 nm/0.23–0.32/-/0.36–0.47%	Y/Epimastigotes and trypomastigotes	After 72 h the NE had an IC50 48 times lower than the free BNZ (0.09 µg/mL and 4.3 µg/mL, respectively).
Liposome	Benznidazole	Free radical formation	2 µm/-/33%/-	RA/Trypopastigote	In vivo results showed that there were no significant differences in the reduction of pasitemia in MV-BNZ and free-BNZ for the strain under study, despite its greater selectivity for liver tissue.
Mesoporous silica nanoparticles	Benznidazole	Free radical formation	100 nm/-/-/-	CL Brener/epimastigote	Encapsulated BNZ showed to be 30-fold better than free BNZ.
Inorganic nanoparticles	Benznidazole	Free radical formation	41.81 nm/-/25%/-	Y/Amastigote, epimastigote and trypomastigote	Nanoparticles showed a 7 and 37-fold lower IC50 than the free-BNZ against epimastigotes and trypomastigote, respectively. On amastigotes, the system also showed great activity with increase in selectivity index.
Solid dispersions	Benznidazole	Free radicals formation	-	Y/Trypomastigote	The suppression of parasitemia was higher in solid dispersions (60%) them free-BNZ (33%).
Solid dispersions	Benznidazole	Free radicals formation	-	Tulahuen/trypomastigote	The trypanocidal efficacy of SD (60 mg/kg/day) was equivalent to commercial BNZ (50 mg/kg/day), but without manifest side effects or hepatotoxicity.
Self-emulsifying delivery system	Benznidazole	Free radical formation	496 nm/-/-/26.5 mg/mL of drug	Y/Amastigote, epimastigote and trypomastigote	The formulation was proven safe at concentrations below 25 µM, nearly three times higher than the IC90 of free BNZ. Anti-*T. cruzi* activity tests demonstrated a dose-dependent inhibition of parasitemia growth, but no difference in activity compared to the free drug. In vivo testing showed similar efficacy and safety results for the SEDDS and free BNZ.
Polymeric nanoparticles	Benznidazole	Free radical formation	63.3/<0.4/98%/-	Nicaragua (TcN)/Amastigote and trypomastigote	NP-BNZ showed a LC50 1.36 and 2-fold lower than free-BNZ in trypostigotes and amastigotes, respectively. The histopathological analysis showed the NP-BNZ decreases cardiac tissue inflammation.
Polymeric microparticles	Benznidazole	Free radical formation	0.87–1.08 μm/-/95–96%/-	Nicaragua (TcN)/trypomastigote	The authors accessed the efficacy of the therapeutic scheme. Both regimens showed good results regarding parasitemia elimination, reduction of *T. cruzi*-specific antibodies and INF-γ-producing cells.
Polymeric nanoparticles	Nifurtimox	Free radical formation	>200/-/33.4%/-	CA-1 (clone)/epimastigote and amastigote	The in vitro results showed higher trypanocidal activity for NP-NFX than free-NFX. That was the first study about Chagas disease treatment and nanotechnology.
Micelles and liposome	Hypericin	Free radical formation	-	Y/Trypomastigote	The studies were performed targeting photodynamic therapy, and the results evaluated with and without light. The polymeric micelles showed efficacy at concentrations higher than 0.8 µM, and EC50 around 0.3–0.4 µM for experiments in the light and 6–8 µM for experiments conducted in the dark.
Polymeric nanoparticles	Allopurinol	Purine salvage inhibition	187 ± 54 nm/−100% EE/62.8 ± 1.9 µg/mg	CA-1 (clones)/Epimastigote	In the in vitro assay, showed higher trypanocidal activity (91.5% at 16.7 µg/mL, IC50 = 0.5 ± 0.1 µg/mL) to NP-allopurinol compared to the free-allopurinol (45.9% at 16.7 µg/mL, IC50 = 37.3 ± 5.0 µg/mL). That is, it presented 74 times lower IC50.
Polymeric nanoparticles	Nitric oxide	Inhibition of cruzipain	270–550 nm/0.35/99%/-	Y/Amastigote, epimastigote and trypomastigote	Dose-dependent activity for both epimastigotes and trypomastigotes in in vitro assay, with IC50 = 252 µg/mL. In vivo assay showed that the concentration of 200 µg/mL had great activity against amastigotes.
Solid lipid nanoparticles	H2bdtc	Inhibition of cruzipain	127 nm/0.22/98.16%/-	Y/Trypomastigote	The studies showed that both the free-H2bdtc and NP- H2bdtc showed similar trypanocidal activity to benznidazole. In the in vivo tests NP-H2bdtc was more effective eliminated 70% of circulating parasites, while free-H2bdtc and BZN eliminated 48 and 15% of parasites, respectively.
Nanoemulsion	Sulfonamides 3F, 3G, 3W, 5B, 5C and 5D	Inhibition of carbonic anhydrase	35–100 nm/PdI <0.28/-/-	Dm28c and Y/epimastigote	All sulfonamides showed IC50 lower than BNZ, but also showed cellular toxicity against RAW 267.4. The best one was the 3F with a IC50 of 3.54 µM. Flow ciitometry evidenced cell death by necrosis in the following proportions of 82.41%, 81.26% and 57.03%, respectively, for the Dm28c strain, being more effective than the reference drug BNZ (effect of 51.16%).
Polymeric nanocapsules	Lychnopholide	Inhibition of trypanothione	LYC-PCL 182.5 nm/-/>95%/- PLA-PEG 105.3 nm/-/100%/-	CL Brener and Y/Trypomastigote	100% cure in animals infected with strain Y treated with LYC-PLA-PEG-NC for 20 days; BNZ and LYC-PCL-NC obtained less than 75% and 62.5% cure, respectively.
Solid dispersion	Ursolic acid	Induction of autophagy	59–115 μm/-/-/-	Y/Trypomastigote	Demonstrated improvements in the solubility and dissolution UA. This explain the improved in vitro trypanocidal activity of ursolic acid in solid dispersion compared to physical mixture.
Polymeric nanoparticles	Ursolic acid	Induction of autophagy	173.2 nm/<0.1/94%/-	Y/Trypomastigote	In vitro study showed no cytotoxicity in LLC-MK2 fibroblasts cells, and the nanoparticles inhibited 56.85 ± 5.7% of trypomastigotes at 30 µM during 24 h of incubation. In vivo assay evidenced a decrease in the parasitemia 3.5-fold better than BNZ.
Nanoemulsion	Ursolic acid	Induction of autophagy	37 nm/<0.2/>87%/-	CL B5 (clone)/amastigote	Nanoemulsion showed to be less effective (IC50 18 µM) compared to BNZ (4.1 µM of), but safer (CC50 396.4 µM) than BNZ (165 µM).
Polymeric microparticles	(−)-Hinokinin	Inhibition of tubulin assembly	862 nm/0.07/72.46/-	CL B5 (clone)/Trypomastigote	NK-loaded microparticles (40 mg/kg, every 2 days) significantly reduced parasitemia compared to untreated controls (*p* < 0.05). HNK-loaded microparticles were more effective than 20 mg/kg/day of free-HNK in reducing parasites (*p* < 0.05).
Nanoarchaeosomes	Imiquimod	Immunomodulation	800 nm/-/-/-	RA/Trypomastigote	The decrease in parasitemia to nanoarq-IMQ was ASC of 213.6 ± 49.7, significantly lower than negative control group (PBS), the blank system and free-IMQ. In the research for specific antibodies, animals treated with nanoarq-IMQ presented four times more IgG1 and IgG2a than animals treated with BNZ, reinforcing the immunomodulatory capacity of IMQ.
Rigid lipid nanosomes	Actinomicin D	Inhibition of RNA synthesis	37 nm/-/-/-	Maracay/epimastigote	The study demonstrated that a 50 ng/mL dose of free-ActD killed 100% of parasites in 24 h. However, when ISCOMs-ActD the same efficacy was achieved with a significantly lower dose of 25.47 × 10^−2^ ng/mL, representing a remarkable 196.8-fold reduction in dosage.
Liposomes	N,N′-squaramide 17	Not elucidated	191.19 ± 59.82 nm/-/73.35%/-	CL Brener/epimastigote and amastigote	The liposomes demonstrated less efficacy than BNZ under the conditions evaluated but showed more cytocompatibility and a greater selective index (48.18 to the system and 14.38 to BNZ).
Silver nanoparticles	Corn cob xylan	Not elucidated	102.3 ± 1.7/-/-/-	Y/epimastigote	NP-xylan at 100 μg/mL significantly reduced by 82% after 24 h and 95% after 48 h of incubation, while free-xylan did not show this effect.

## Data Availability

No new data were created or analyzed in this study. Data sharing is not applicable to this article.

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
