# Peer review of "From Benznidazole to New Drugs: Nanotechnology Contribution in Chagas Disease"

_ijms, 2023, doi:10.3390/ijms241813778_

Round 1

Reviewer 1 Report

The authors wrote a review on the use of nanotechnology as a new attempt to treat chronic patients with Chagas disease, which occurs in many patients mainly in South America. 1) Chagas disease, 2) Nanotechnology, 3) Targets of treatment, and the actual research results are well covered. Reviewers are impressed by the very easy-to-understand review.

Minor editing of English language required.

Author Response

Natal-RN Brazil, July 29, 2023

Dear Editors,

We would like to thank the Editors and referees for the careful analysis of our paper and constructive suggestions. We believe that the remarks helped us to improve the message and overall clarity of the paper. We believe that we have attended to the suggestions and changes in the revised version. Please find below our responses to reviewers’ comments and the list of changes made in RED in the original text. In addition, the revised version of the manuscript has been edited for spelling and grammar.

RESPONSE TO THE COMMENTS OF THE REVIEWER #1

General comment:

- The authors wrote a review on the use of nanotechnology as a new attempt to treat chronic patients with Chagas disease, which occurs in many patients mainly in South America. 1) Chagas disease, 2) Nanotechnology, 3) Targets of treatment, and the actual research results are well covered. Reviewers are impressed by the very easy-to-understand review.

Response: We appreciate the referee’s comment and positive feedback about the overall message of the manuscript. In addition, we revised the entire manuscript to improve the explanation about our proposal accordingly. Please find below the responses to the comments and an itemized list of changes in the manuscript highlighted in RED.

Specific Comments on the quality of english language: “Minor editing of English language required.”

Response: The revised version of the manuscript has been edited for spelling and grammar. In addition, please find the substitutions:

Benznidazol for Benznidazole (in Title); Eficcacy for efficacy (in line 53); Teratment for treatment (in line 73); composunds for compounds (in line 51).

Reviewer 2 Report

The paper proposes to review nanotechnologies and their suitability to develop new therapeutics against T. cruzi infection. It is an ambitious and valuable aim! Unfortunately, it is not structured as a systematic review and the narrative form of the review makes it very difficult to read and follow. The data extracted from a paper are given in extenso and unfortunately, no tentative to give a synthetic view in the framework of T. cruzi chemotherapy is performed. I suggest adding numerical data extracted from the literature in a table gathering information on dosage, carrier molecules, model parasite stage, toxicity etc.. and discussing them in a related chapter. A final discussion would be also of great interest to the reader. This has the potential to interest researchers working in the field of parasitology, experimental and applied pharmacology but also to medics and healthcare professionals. 

Remarks.

Lines 145 to 166 these paragraphs do not relate to a "General aspect of nanocarriers".

Chap 2. Chagas disease. More information is required on physiopathology, treatment, and specificity of Chagas disease treatment.

Line 110. The sentence mean that more parasitic stage stages are present, if yes please describe them.

Line 120-125 All these assertions need to be carefully discussed because some are not widely accepted.

Line 136 "relevant"? not "debilitating"

Line 167. They are the most frequently used, for what in which context against what?

Minor

Table 1 first column, I guess it is the patent name that is given and not the "title of the product".

Italicise species names T. cruzi etc.

The review as it stands is very difficult to follow due to its absence of a synthetic view and discussion of the information given in the manuscript.

Author Response

Natal-RN Brazil, July 29, 2023

Dear Editors,

We would like to thank the Editors and referees for the careful analysis of our paper and constructive suggestions. We believe that the remarks helped us to improve the message and overall clarity of the paper. We believe that we have attended to the suggestions and changes in the revised version. Please find below our responses to reviewers’ comments and the list of changes made in RED in the original text. In addition, the revised version of the manuscript has been edited for spelling and grammar.

RESPONSE TO THE COMMENTS OF THE REVIEWER #2

General comment:

- The paper proposes to review nanotechnologies and their suitability to develop new therapeutics against T. cruzi infection. It is an ambitious and valuable aim! Unfortunately, it is not structured as a systematic review and the narrative form of the review makes it very difficult to read and follow. The data extracted from a paper are given in extenso and unfortunately, no tentative to give a synthetic view in the framework of T. cruzi chemotherapy is performed. I suggest adding numerical data extracted from the literature in a table gathering information on dosage, carrier molecules, model parasite stage, toxicity etc.. and discussing them in a related chapter. A final discussion would be also of great interest to the reader. This has the potential to interest researchers working in the field of parasitology, experimental and applied pharmacology but also to medics and healthcare professionals.

Response: We appreciate the referee’s comment. We believe that all the suggestions contributed to improving the clarity and quality of the message in the manuscript. Please find the responses and altered text highlighted in RED.

MAJOR COMMENT 1.

            In this review, we investigate and discuss the results obtained in in vitro and in vivo studies regarding the trypanocidal effect of several drugs incorporated into different nanosystems for the treatment of Chagas disease.

As suggested, a new Table 2 was added in the text. Please, find grouped the studied drugs according to their cellular target in Trypanosoma cruzi, such as drugs that act on ergosterol, free radical formation, enzyme inhibition (cruzipain, carbonic anhydrase, trypanothione synthetase), induction of autophagy, inhibition of tubulin assembly, immunomodulation, inhibition of RNA synthesis and others, as shown in Table 2.

MAJOR COMMENT 2:

A new synthetic discussion topic has been added to the text, please find as described: “4.9. Discussion

Specific comment 1: Lines 145 to 166 these paragraphs do not relate to a "General aspect of nanocarriers".

Response: We understand and appreciate the reviewer comment. The text was changed accordingly. This issue is addressed in the following topics as strategies for targeting different molecules for the treatment of Trypanosoma cruzi infection. The lines 155-169 introduce chapter 3.

Specific comment 2: Chap 2. Chagas disease. More information is required on physiopathology, treatment, and specificity of Chagas disease treatment.

Response: The text was rewritten and suggested information addressed. Please find changed text in RED, in lines 120-169.

Specific comment 3: Line 110. The sentence mean that more parasitic stage stages are present, if yes please describe them.

Response: The sentence was improved and clarified. Please checked the changed text in lines 110-111.

Specific comment 4: Line 120-125 All these assertions need to be carefully discussed because some are not widely accepted.

Response: The text was rewritten and accordingly cited. Please find lines 120-126.

Specific comment 5: Line 136 "relevant"? not "debilitating"

Response: bring our attention to this. The most coherent term is debilitating. The sentence has been rewritten, and the appropriate term employed to “Chronic Chagas cardiomyopathy is one of the most debilitating clinical forms of the disease, affecting about one third of infected individuals [36,37]. This condition can cause serious consequences such as sudden death, cardiac arrhythmias, heart failure and thromboembolism [38]. Therefore, CD is considered a disabling disease and re-sponsible for the highest morbidity and mortality among parasitic diseases [39], but the mechanisms or determinants responsible for the development of cardiomyopathy are nuclear [40].” Please also check the line 148.

Specific comment 6: Line 167. They are the most frequently used, for what in which context against what?

Response: The clarity of text was improved. Please check the lines 168-169.

Specific comment 7: Table 1 first column, I guess it is the patent name that is given and not the "title of the product".

Response: The first column was changed accordingly.Please check Table 1, line 92.

Specific comment 8: Italicise species names T. cruzi etc.

Response: The text has been revised and all appropriate terms have been italicized. Please also check the corresponding lines. “T. cruzi” in line 45; “T. cruzi” in line 50; “T. cruzi” in line 56; “Trypanosoma cruzi” in line 63; “Trypanosoma cruzi” in line 268; “ Trypanosoma cruzi” in line 273; “T. cruzi” in line 278; “T. cruzi” in line 282; “T. cruzi” in line 286; “T. cruzi” in line 300; “T. cruzi” in line 319; “T. cruzi” in line 323; “T. cruzi” in line 330; “T. cruzi” in line 337; “T. cruzi” in line 367; “T. cruzi” in line 375; “T. cruzi” in line 376; “T. cruzi” in line 383; “T. cruzi” in line 405; “T. cruzi” in line 415; “T. cruzi” in line 429; “T. cruzi” in line 439; “T. cruzi” in line 451; “T. cruzi” in line 466; “T. cruzi” in line 446; “T. cruzi” in line 473; “T. cruzi” in line 480; “T. cruzi” in line 491; “T. cruzi” in line 515; “Hypericum perforatum” in line 528; “T. cruzi” in line 529; “T. cruzi” in line 534; “T. cruzi” in line 557; “T. cruzi” in line 568; “T. cruzi” in line 575; “T. cruzi” in line 578; “T. cruzi” in line 636; “T. cruzi” in line 623; “T. cruzi” in line 650; “Eugenia caryophyllus” in line 652; “T. cruzi” in line 654; “T. cruzi” in line 664; “T. cruzi” in line 668; ”Lychnophora trichocarpha” in line 668; “T. cruzi” in line 687; “T. cruzi” in line 692; “T. cruzi” in line 695; “T. cruzi” in line 701; “T. cruzi” in line 728; “T. cruzi” in line 731; “T. cruzi” in line 750; “Halorubrum tebenquichense” in line 766; “T. cruzi” in line 767; “Streptomyces” in line 782; “T. cruzi” in line 784; “T. cruzi” in line 789; “T. cruzi” in line 794; “Trypanosoma cruzi” in line 891.

Reviewer 3 Report

After reading the manuscript “From benznidazol to new drugs: Nanotechnology contribution in Chagas Disease”, I have found it is an interesting manuscript.

 As indicated by the authors, new effective treatments for Chagas disease are far from being achieved. The search for new formulations that increase the efficacy or reduce the toxicity of known active compounds is a very interesting approach.

However, prior to further processing of the paper several points need to be clarified.

 -In my opinion Figure 2 does not contribute much to the study and can be eliminated or reduced in size

- Why do the authors only include patents from Brazil? I think it would be more interesting to include patents (from any country) that have been used in clinical trials.

-Please check that the names of genus and species are in italics

Author Response

Natal-RN Brazil, July 29, 2023

Dear Editors,

We would like to thank the Editors and referees for the careful analysis of our paper and constructive suggestions. We believe that the remarks helped us to improve the message and overall clarity of the paper. We believe that we have attended to the suggestions and changes in the revised version. Please find below our responses to reviewers’ comments and the list of changes made in RED in the original text. In addition, the revised version of the manuscript has been edited for spelling and grammar.

RESPONSE TO THE COMMENTS OF THE REVIEWER #3

General comment:

- After reading the manuscript “From benznidazol to new drugs: Nanotechnology contribution in Chagas Disease”, I have found it is an interesting manuscript. As indicated by the authors, new effective treatments for Chagas disease are far from being achieved. The search for new formulations that increase the efficacy or reduce the toxicity of known active compounds is a very interesting approach. However, prior to further processing of the paper several points need to be clarified.

Response: We appreciate the referee’s comments and positive feedback. We believe that all the suggestions contributed to improving the clarity and quality of the message in the manuscript. Please find the responses and altered text highlighted in RED.

Specific comment 1: In my opinion Figure 2 does not contribute much to the study and can be eliminated or reduced in size.

Response: Thank you for your suggestion. Figure 2 refers to a word cloud, which is a graph that shows the frequency with words or terms appear in the used references, and are generated automatically. As suggested, This Figure have the size reduced. Please check Figure 2, line 89.

Specific comment 2: Why do the authors only include patents from Brazil? I think it would be more interesting to include patents (from any country) that have been used in clinical trials.

Response: A new search for registered patents on nanosystems as drug delivery systems for the treatment of Chagas disease was carried out and a new patent was added, again from Brazil with also international registration. The patent databases consulted were Espacenet, Google Patents and INPI (Brazilian patent database). It is our knowledge that new drugs are in the clinical trial phase and that there are patent registrations of new drugs for the treatment of the disease. However, most of them do not have nanotechnology applied to the drugs. An important fact is that most of the articles found in our search and cited in our review (53%) are from Brazilian research groups (please check the 68 line), which justifies that we found with our searches only patents from Brazil. The added patent was “Compositions pharmaceutical and veterinary in Micro- and nanostructured forms and forming micro- and nanostructures in the gastrointestinal tract containing benznidazole and its derivatives and their Biological applications”. Please also check the Table 1, line 92.

Specific comment 3: Please check that the names of genus and species are in italics

Response: The text has been revised and all appropriate terms have been italicized. Please also check the corresponding lines. “T. cruzi” in line 45; “T. cruzi” in line 50; “T. cruzi” in line 56; “Trypanosoma cruzi” in line 63; “Trypanosoma cruzi” in line 268; “ Trypanosoma cruzi” in line 273; “T. cruzi” in line 278; “T. cruzi” in line 282; “T. cruzi” in line 286; “T. cruzi” in line 300; “T. cruzi” in line 319; “T. cruzi” in line 323; “T. cruzi” in line 330; “T. cruzi” in line 337; “T. cruzi” in line 367; “T. cruzi” in line 375; “T. cruzi” in line 376; “T. cruzi” in line 383; “T. cruzi” in line 405; “T. cruzi” in line 415; “T. cruzi” in line 429; “T. cruzi” in line 439; “T. cruzi” in line 451; “T. cruzi” in line 466; “T. cruzi” in line 446; “T. cruzi” in line 473; “T. cruzi” in line 480; “T. cruzi” in line 491; “T. cruzi” in line 515; “Hypericum perforatum” in line 528; “T. cruzi” in line 529; “T. cruzi” in line 534; “T. cruzi” in line 557; “T. cruzi” in line 568; “T. cruzi” in line 575; “T. cruzi” in line 578; “T. cruzi” in line 636; “T. cruzi” in line 623; “T. cruzi” in line 650; “Eugenia caryophyllus” in line 652; “T. cruzi” in line 654; “T. cruzi” in line 664; “T. cruzi” in line 668; ”Lychnophora trichocarpha” in line 668; “T. cruzi” in line 687; “T. cruzi” in line 692; “T. cruzi” in line 695; “T. cruzi” in line 701; “T. cruzi” in line 728; “T. cruzi” in line 731; “T. cruzi” in line 750; “Halorubrum tebenquichense” in line 766; “T. cruzi” in line 767; “Streptomyces” in line 782; “T. cruzi” in line 784; “T. cruzi” in line 789; “T. cruzi” in line 794; “Trypanosoma cruzi” in line 891